# Transcriptome and Methylome Analysis Reveal Complex Cross-Talks between Thyroid Hormone and Glucocorticoid Signaling at Xenopus Metamorphosis

**DOI:** 10.3390/cells10092375

**Published:** 2021-09-09

**Authors:** Nicolas Buisine, Alexis Grimaldi, Vincent Jonchere, Muriel Rigolet, Corinne Blugeon, Juliette Hamroune, Laurent Marc Sachs

**Affiliations:** 1UMR7221 Molecular Physiology and Adaption, CNRS, Museum National d’Histoire Naturelle, 57 Rue Cuvier, CEDEX 05, 75231 Paris, France; buisine@mnhn.fr (N.B.); alexis.grimaldi@gmail.com (A.G.); vincent.jonchere@inrae.fr (V.J.); muriel.rigolet@mnhn.fr (M.R.); 2Genomics Core Facility, Département de Biologie, Institut de Biologie de l’ENS (IBENS), École Normale Supérieure, CNRS, INSERM, Université PSL, 75005 Paris, France; blugeon@biologie.ens.fr (C.B.); juliette.hamroune@inserm.fr (J.H.)

**Keywords:** Xenopus metamorphosis, thyroid hormone, glucocorticoids, cross-talks, functional genomics, DNA methylation

## Abstract

Background: Most work in endocrinology focus on the action of a single hormone, and very little on the cross-talks between two hormones. Here we characterize the nature of interactions between thyroid hormone and glucocorticoid signaling during *Xenopus tropicalis* metamorphosis. Methods: We used functional genomics to derive genome wide profiles of methylated DNA and measured changes of gene expression after hormonal treatments of a highly responsive tissue, tailfin. Clustering classified the data into four types of biological responses, and biological networks were modeled by system biology. Results: We found that gene expression is mostly regulated by either T_3_ or CORT, or their additive effect when they both regulate the same genes. A small but non-negligible fraction of genes (12%) displayed non-trivial regulations indicative of complex interactions between the signaling pathways. Strikingly, DNA methylation changes display the opposite and are dominated by cross-talks. Conclusion: Cross-talks between thyroid hormones and glucocorticoids are more complex than initially envisioned and are not limited to the simple addition of their individual effects, a statement that can be summarized with the pseudo-equation: TH **∙** GC > TH + GC. DNA methylation changes are highly dynamic and buffered from genome expression.

## 1. Introduction

Thyroid hormones (TH) and Glucocorticoids (GC) are ubiquitous mediators of endocrine signalling systems coordinating homeostasis, a response to environmental challenges and development throughout life, starting from early development until death. These systems are highly conserved in vertebrates [1,2,3], where they regulate similar processes across taxa ranging from fish to amphibians to birds and mammals [4]. Their action is very diverse [5,6,7,8,9,10,11,12,13], and both signalling pathways often cooperate in biological processes (e.g., during bone growth and differentiation [14], brain maturation [15,16], and liver metabolism [17]), where affecting one or the other signalling pathway results in various pathologies and developmental defects [5,18,19]. This is at the very heart of the biological question we ask, and we will list below some molecular details to highlight some sources of functional interactions between pathways.

The action mechanism of these hormones involves specific receptors belonging to the super family of nuclear receptors (NR) transcription factors [20]. They directly regulate the expression of a number of target genes, typically in the order of a few thousands [21,22,23]. Transcriptional regulation may also be indirect, as direct target encoding transcription factors may induce a secondary wave of transcriptional changes [24,25]. Importantly, nuclear receptors have a natural tendency to functionally interact with each other, thus making their corresponding pathways cross-dependent of one another [26]. For instance, this is the case for GC and estrogen signalling [27,28], TH and estrogens [27,29,30,31], and GC and androgens [32]. At the mechanistic level, NRs can synergistically bind or compete to DNA at shared response elements [33,34].

TH and GC action is not only dependent on the receptor binding to DNA (for TH [35,36]; for GC [37]). This involves direct action of the hormone at other targets, or a cytoplasmic action of the hormone bound receptor. Even cytosolic factors are potential platforms for functional interactions between pathways, as exemplified by PI3K, which physically interacts with both THR and GR [38,39]. Despite much evidence that TH and GC pathways functionally interact, very little details are known about the mechanisms involved and their general properties.

In this paper, we provide a detailed description of the nature of the functional interactions between TH and GC signaling. Our working model is Xenopus metamorphosis because TH is necessary and sufficient to initiate metamorphosis [40,41], while CORT acts synergistically with TH to accelerate progression of TH-induced metamorphosis [42,43,44,45] and is essential for survival at the climax [46]. Anuran metamorphosis marks the end of larval developments and coincides with the transition from a water-based to an air-breathing life style and anatomy [41,47]. This process is very fast (<two weeks) and controllable in laboratory settings. Remarkably, organotypic culture of tail explants fully recapitulate tail regression in vitro, without the confounding effects of body-level feedback regulation loops [48]. Here, we choose to focus on tailfin because (1) tail tissue is highly responsive to TH and GC [44,49], and (2) tailfin displays a limited diversity of cell types.

Previous reports [50,51] demonstrated that GC regulates the expression of *dio2* and *dio3* genes, which encode enzymes metabolizing THs and regulate the availability of the biologically active hormone (T_3_). Through this action, GC increases the activity of the DIO2 enzyme, resulting in enhanced transition from T_4_ into biologically more active T_3_, and decreases the activity of DIO3, responsible for the degradation of T_3_ into (less) inactive products. In a seminal work, Kulkarni demonstrated that co-treatment of pre-metamorphic tadpoles results in unexpectedly complex biological responses [52].

In this work, we used a combination of hormone treatment on the highly TH responsive tailfin tissue, on whole animals and explant cultures. We measured variations of gene expression by RNA-Seq followed by in-depth modelization of biological signals with system biology technologies. Overall, we demonstrate that the action of TH and GC is not limited to the simple addition of the effects of TH and GC, and that the known action of GC on *dio2/dio3* cannot explain the diversity of transcriptional responses in tailfin. In other words, we identified many novel components of TH and GC interactions. From therein, we will refer to the complete set of functional interactions between pathways with the terms “cross-talks” or “X-talks”, and this accounts for already known mechanisms of action (synergy, cooperation) and well as any novel mechanism of action. At the transcriptional level, we demonstrate that, quantitatively, TH and GC effects are mostly independent or additive (i.e., the independent action of both hormones), and that cross-talks are nonetheless relatively frequent and display a large diversity of biological responses. Surprisingly, we found a strong transcriptional reprograming of the DNA-methylation machinery. We therefore profiled changes of DNA methylation levels genome-wide. Contrary to our expectations, we found that, quantitatively, the DNA methylation dynamic is dominated by the complex interactions between TH and GC, in complete opposition to the transcriptional response.

We thus propose a new picture of the interactions of TH and GC, and between two hormones in broader terms, which is far more complex than initially realized.

## 2. Methods

### 2.1. Animal Care

*Xenopus tropicalis* (*X. tropicalis*) tadpoles were obtained from the Centre de Ressources Biologiques (Rennes, France), raised at 26 °C in dechlorinated tap water, and fed with nettle powder. Developmental stages were set according to the normal table of *Xenopus laevis* (Daudin) [53]. Animal care was in accordance with institutional and national guidelines (ref: 68008, delivered by the Cuvier Ethic Committee).

### 2.2. Whole Tadpole Hormonal Treatments

The most biologically active TH, 3,3′,5′-triiodothyronine (T_3_, T2752, SIGMA, Lezennes, France) was dissolved in 0.1 N NaOH and added to the culture medium or the tank to a final concentration of 10 nM. Corticosterone (CORT, C2505, SIGMA) was dissolved in 100% DMSO (D8418, SIGMA), and added to the culture medium or the tank to a final concentration of 100 nM. All treatments received an equivalent amount of DMSO vehicle (0.001%). For whole tadpole hormonal treatment, 5 tadpoles at stage NF-53-54 were placed in a 1 L beaker containing 500 mL of dechlorinated tap water, where the hormones have been previously added. For transcriptome and DNA methylome analysis, tadpoles were euthanized 24 h later with an overdose of anesthesia (0.01% MS222, SIGMA), prior to dissection of tailfin skin.

### 2.3. Organotypic Tail Culture

Th tail of stage NF53-54 *X. tropicalis* tadpoles were amputated just above the posterior legs. The tails were then dipped in 100% ethanol and washed with 65% L15 (11415-049, GIBCO, France) + antibiotic/antimycotic (15240-96, GIBCO). Each tail was then cultured in 24 well culture plates (TPP), with 1 mL 65% L15 + antibiotic/antimycotic and T_3_ and/or CORT at 24 °C, protected from light. After 24 h, tailfin skin was dissected from the whole tail, snap frozen in liquid nitrogen, and stored at −80 °C. Three independent biological replicates were used for the RNA-Seq and another set of >8 independent replicates were used for RT-qPCR validations.

### 2.4. RNA Isolation and Measure of Gene Expression

Tissues from either cultured tail explants or whole tadpoles were processed as described in [54]. RNAs were quantified with a NanoDrop spectrofluorometer and their quality controlled with Agilent RNA 6000 nano chips on a Bioanalyzer before treatment with DNAse (TURBODNAse, Ambion, UK). Reverse transcription of mRNA was carried out with SUPERSCRIPT v3 following the manufacturer’s recomendations and subject to quantification of cDNA abundance by conventional RT-qPCR. The endogenous control rpl8 was selected based on NormFinder [55] analysis of a panel of candidate genes. Raw results were processed using the −2^ΔΔCt^ method. Data were normalized on the endogenous control rpl8 (ΔCt). For each treatment (T_3_, CORT, T_3_ + CORT), ΔCt were normalized on the non-treated control. Resulting values correspond to the expression fold-change compared to the non-treated control in log2 scale. Statistical significance was addressed with a Mann-Waitney test. Primer sequence is described in Appendix A.

Library preparation and Illumina sequencing were performed at the Paris Genomic Center (France). Messenger (polyA+) RNAs were purified from 1 μg of total RNA using oligo(dT). Libraries were prepared using the strand non-specific RNA-Seq library preparation TruSeq RNA Sample Prep v2 kit (Illumina, France). A 50 bp single read sequencing was performed on a HiSeq 1500 device. A mean of 67 ± 5 million passing Illumina quality filter reads was obtained for each of the 32 samples. Read qualities were assessed with the FASTQC toolkit v0.11.3 (http://www.bioinformatics.babraham.ac.uk/projects/fastqc/).

### 2.5. RNA-Seq Data Processing

Redundant reads were filtered by keeping the read with the best quality score. We used the *fastx* toolkit (v 0.0.13) to clip the 3′ end of reads when the score dropped below 30 on the Sanger scale (Phred + 33). Preprocessed reads were mapped on the version 4.1 of the *X. tropicalis* genome [56] using bowtie 0.12.3 [57] with the following parameters: “-5 10 -m1 -n2 -l28”. Gene expression call is based on models aggregated from Ensembl, Xenbase, and [58]. Overall, uniquely mapped read mapping efficiency was higher than 75%. Redundancy removal further reduced the mapped read count by a factor of two, resulting in a uniquely mapped and non-redundant read count ≥ 22.10^6^ (not shown). Consistency between replicates and treatments was assessed by Principal Component Analysis (PCA): raw read counts were subjected to a variance-stabilization transformation as described in [59]. Differential expression between treatments was performed with DESeq [59] version 1.12 with the following parameters: method = “pooled”, sharing-mode = “maximum”, and fit-type = “parametric”. Genes with low expression values were discarded as described in the DESeq, with *θ* = 0.4. Genes statistically differentially expressed were called at an FDR of 5%. Genes were grouped in 81 clusters according to their expression profiles.

Culture effects were removed by filtering out genes without similar expression profiles in both cultured tail explants and whole tadpoles. Only genes DE with a fold change higher or equal to two-fold changes were considered in the whole tadpole data sets.

Gene ontology analysis is based on GORILLA software suite [60]. Most significant categories are shown either as a bar graph or a circle plot. In this case, the size of each circle is set by the number of genes in each category, and color is inversely proportional to the *p*-value of the term enrichment.

### 2.6. Clustering

Clustering of DE genes is aimed at classify individual genes into a number of a specific “response types”. Expression values of each gene across the four treatment conditions (CTRL, T_3_, CORT, and T_3_-CORT) were standardized by setting their average to 0 and their variance to 1. For each treatment, the normalized gene expression level is compared to CTRL and used to derive whether it is up- (‘u’), down- (‘d’) or not- (‘n’) regulated after each treatment. Genes are then assigned to a cluster named after the corresponding letters arranged in the T_3_, CORT, and T_3_-CORT order. This compact notation summarizes transcriptional responses. For example, gene transcription only induced with T_3_ is labeled u_n_u: transcription is up with T_3_ (first ‘u’), not affected with CORT (middle ‘n’), and up after T_3_-CORT co-treatment (last ‘u’). Similarly, CORT only responsive genes are n_u_u or n_d_d, and genes transcription regulated by both T_3_ and CORT belong to d_d_d or u_u_u. Non trivial regulations (i.e., X-talks) also become explicit. For instance, u_n_n corresponds to a transcription level up in T_3_, but with no change after CORT and T_3_-CORT treatments. In this case, despite no direct action on its own, CORT cancels the action of T_3_. The threshold used to call u and d is set as the smallest for which the n_n_n cluster contains no gene (i.e., zero genes that are not DE at least once). This very simple mathematical transformation has three strong advantages:-This is a gene-level transformation, with few constraints on the actual number of genes within each cluster. This differs from the widely used k-mean clustering, which tends to produce clusters with similar number of genes. It is clear that there is no reason a priori to impose that each type of biological response should be constrained in term of the number of genes.-It has better control of false negatives because it does not call multiple times for statistics with limited power, as calling multiple times for differential analysis performed with few biological replicates (*n* = 3, as the current standard suggests) would. Thus, statistics may deviate slightly from the differential analysis.-The biological response of genes can be compared without the confounding effect of their expression level, which range over five orders of magnitude.

### 2.7. Signaling and Metabolic Network

In this work, we focus on two types of biological pathways: signaling and metabolism for modeling biological processes, and protein-protein interactions for modeling molecular mechanisms. The information needed for building each network are available at dedicated databases: the KEGG pathways database (Kyoto Encyclopedia of genes and Genomes database), which mostly focuses on signaling and metabolism, for biological processes, and BIOGRID, which is a repository of known protein-protein interactions, for modeling molecular mechanisms.

The signaling-metabolic network is built upon pathways extracted from the KEGG pathways database with the JEPETTO plugin [61] of the Cytoscape v3.8.2 environment [62]. All KEGG pathways containing at least one DE gene were collected and merged to create a network, where nodes correspond to gene products and links describe the functional connections between them. This step is easily carried out by populating a square matrix representation of the network. Network properties were computed with in-house scripts, in a manner similar to other published tools (e.g., networkX). Network lay-out computed with the “edge weighted spring-embedded” algorithm. Hubs are defined as nodes (i.e., gene product) with a degree (a.k.a. connectivity) higher than 20. Fitting the degree distribution to a power low function confirms that the network is scale free and displays small world properties (not shown).

### 2.8. PPI Network

Protein-protein interactions (PPI) were from BIOGRID v 4.4.198 [63], downloaded as a CSV dump file. PPI were extracted and formatted with standard Unix tools (grep and gawk) before being loaded into CYTOSCAPE. The gene content of metabolic pathways and apoptosis were from the KEGG pathway database [64]. Fitting a power low function on degree distribution demonstrates that the reconstructed network is scale free and displays small world properties (not shown).

### 2.9. Prediction of Nuclear Receptors Binding Sites

Nuclear receptors binding sites are notoriously difficult to detect because they are composed of two half sites in direct, everted, or inverted orientation, separated from each other by a spacer of a length of 0 to 8 bp [65,66]. The diversity of binding sites topology prevents the direct use of position-specific scoring matrices. Instead, the NHR-scan software [67] relies on a Hidden Markov Model that explicitly models the various topologies. Unfortunately, the published model contains many errors, as the sum of all probabilities of several nodes did not sum up to one, thus making the design improper for probabilistic modeling. Many state transitions were also missing. We corrected the model by adding missing links and setting equal probability between novel links reaching each state, so that the total sum of probabilities at each state equals to 1.

### 2.10. MethylCap-Seq and Identification of Differentially Methylated Regions (DMRs)

Purification of methylated DNA was carried out by affinity columns with methyl-DNA binding proteins (MethylCap Kit), following the manufacturer’s protocol (Diagenode; Denville, NJ, USA). DNA was eluted with buffers of increasing ionic strength, and only the fractions corresponding to moderate to high levels of methylation were kept. Library preparation and Illumina sequencing were performed at the Paris Genomic Center (France). Libraries were prepared using NEXTflex ChIP-Seq Kit (Bioo Scientific), using 30 ng of purified genomic DNA. Libraries were multiplexed by 8 on 2 flowcell lanes. A 50 bp read sequencing was performed on a HiSeq 1500 device (Illumina). A mean of 209 ± 20 million passing Illumina quality filter reads was obtained for each of the 16 samples.

Read mapping was run with bowtie at highest stringency (parameters −l 50, −n 1, −m 1, −5 0 −3 0), producing a total of 74 to 95 million uniquely mapped and non-redundant reads per sample. Read density profiles were generated for each treatment condition and signal intensity was normalized for sequencing depth. Variations of DNA methylation relative to the control were derived by subtracting the CTRL density profile from each of the T_3_, CORT, and T_3_-CORT profiles. The signal of this differential profile was smoothed by using the average value computed over a 10 bp sliding window. Peaks correspond to either an increase or decrease of DNA methylation levels. For the sake of computation time, the whole process was implemented in C and automated in a custom shell script. This resulted in a ~10,000× speed up compared to the same programs written in PYTHON.

In itself, the detection of DMRs is based on non-parametric permutation tests aimed at finding the maximal width and peak height obtained when comparing two profiles without biological contrast. This defines the thresholds of biological and technological noise above which peaks correspond to actual DMRs, with their width and height corresponding to the genome span of the DMR, and the amplitude of DNA methylation changes, respectively. By nature, this DMR detection strategy is fairly conservative, which would limit background noise.

The pipeline works as follows: we created two datasets of reads randomly sampled from the combined pool of all MethylCap-Seq reads (i.e., without biological contrast), before generating the corresponding differential profile and scoring for the longest and highest peaks. This process was iterated 100 times and the highest width and height were kept. Quantitative data are visualized with the JBROWSE genome browser [68].

DMR have then been classified in various response types by following the same procedure as for RNA-Seq (clustering). DMR nomenclature follows the same convention.

## 3. Results

### 3.1. Experimental System to Address Tailfin Regression

The experimental system is the following (Figure 1): pre-metamorphic tadpoles were treated with either 10 nM T_3_, 100 nM CORT, or both, and tailfins were collected before measuring the quantitative variations of RNA and methylation levels (by RNA-Seq and MethyCap-Seq, respectively). To differentiate between autonomous changes of tailfin from the effects of feedback loops originating from the central nervous system, hormone treatments were applied to organotypic tail cultures, although this may introduce confounding effects of the culture process itself. To filter them out, whole animals were also treated, and only the genes found consistently regulated in both datasets were kept. Downstream processing and analysis follow state of the art functional genomics workflows coupled to system biology, as well as the use of novel bio-informatic tools.

### 3.2. Standard Analysis Suggests Additive T_3_ and CORT Effects

RNA abundance per gene was measured by RNA-Seq on tailfins originating from cultured tail explants, following standard protocols and conventional bio-informatic pipelines. We first subjected the reads-count tables to principal component analysis, which aims at controlling the correspondence between components of the total biological variability captured by RNA-Seq and the effects of hormone treatments (Figure 2A). We found that almost 80% of the total variance is projected in the first two principal components, which capture the transcriptional changes induced in response to T_3_ (PC1, 48% of the total variance) and CORT (PC2, 29% of the total variance) treatments. This is indicative of strong biological responses, high contrast between treatments, and very little technical and/or biological noise. As a technical validation of the RNA-Seq procedure, we performed independent measures of gene expression changes by RT-qPCR, on a number of genes displaying various amplitudes of transcriptional response (Figure 2B). Results show strong correlation between expected (RNA-Seq) versus observed (RT-qPCR) signals, illustrating the robustness of our measures of gene expression by RNA-Seq. As expected [69], *thbzip* and *thrb* genes expression is strongly induced after treatment with T_3_ and T_3_-CORT (Appendix A).

Overall, in tailfin culture, the total number of differentially expressed (DE) genes is relatively modest: 303 DE genes with T_3_, 342 with CORT, and 1163 with T_3_-CORT. This represents a non-redundant set of 1363 DE genes. Filtering out culture effects further reduces the number of DE genes down to 186 for T_3_, 114 for CORT, and 655 for T_3_-CORT (Figure 2C and Appendix A). This corresponds to a final non redundant set of 729 DE genes. Despite a limited number of DE genes following treatment with either T_3_ or CORT alone, the number of DE genes after co-treatment is about four times higher (655 vs. 186 and 114), indicative of a strong and specific transcriptional reprogramming.

Gene ontology (GO) analysis is often used to summarize the functional categories of the transcriptional response. Despite limited sensitivity, it is useful to highlight the few dominant biological processes acting in the biological responses (Figure 2D, Appendix A). As expected in the context of tail regression, T_3_ responsive genes involve the metabolism of various extracellular matrix components. On the other hand, CORT responsive genes relate to the metabolism of various small molecules (alcohol, sulfur, and detoxification). The most significant biological processes after the T_3_ -CORT co-treatment correspond to a combination of the terms found after T_3_ (extracellular matrix) or CORT (small molecules metabolism) treatments, and very little terms relative to known TH biology and developmental processes. Overall, this very common type of analysis suggests that the biological output of a T_3_-CORT co-treatment would be as simple as the addition of the effect of each hormone alone. As we will detail below, this view is very broad and partly results from a sub-optimal (albeit classic) analysis design.

### 3.3. Cross-Talks Do Exist, and They Only Represent a Fraction of Transcriptional Responses

The next part of the analysis is aimed at clustering genes based on their expression patterns. Clusters are named with a three letter code depending on whether the (normalized) expression level is higher (‘u’ for up), lower (‘d’ for down), or not different (‘n’) from the control. The three letters are arranged in the following order: T_3_, CORT, and T_3_-CORT. For example, the cluster d_n_d corresponds to the set of genes which are down-regulated after treatment with T_3_ and T_3_-CORT, but unaffected by CORT alone, and the cluster n_u_u contains genes are up-regulated after treatment with CORT and T_3_-CORT, but unaffected by T_3_ alone. These two examples actually correspond to the simple scenario where gene expression is affected by one hormone only, with little or no interaction between them (‘T_3_’ and ‘CORT’). A more complex scenario is ‘ADDITIVE’ (clusters u_u_u and d_d_d), where gene expression is induced or repressed by both hormones individually and the resulting expression level corresponds to the addition of their effect. The third scenario, ‘X-talks’, corresponds to all other cases with functional interactions between TH and GC pathways.

The overall proportions of each categories are shown Figure 3A. A majority of genes belong to ADDITIVE (44.5%, 376/845, Figure 3B), where gene expression is regulated by T_3_ and CORT, and T_3_, where genes only respond to T_3_ (38.8%, 328/845, Figure 3C). Very little genes belong to the CORT category (4.5%, 38/845, Figure 3C). Surprisingly, a non-negligible fraction of genes (12.2%, 103/845) display alternative biological responses which neither respond to T_3_, CORT or ADDITIVE. We collectively refer to this family of responses as “X-talks” (Figure 3D). This category is composed of multiple clusters, where one hormone cancels the action of the other (clusters d_n_n, n_u_n, u_d_n, u_n_n), where the action of one hormone dominates over the other (d_u_d, u_d_u). Other cases are more complex; for example, the u_n_d cluster corresponds to genes where CORT action, which has no effect of its own, and results in a response opposite to that of T_3_ alone. The cluster n_u_d is a similar case, but with T_3_ modulating the CORT response. These clusters are usually composed of only a single gene. Finally, two clusters (36.9% of X-talks genes, 38/103) are particularly notable: n_n_u and n_n_d (Figure 3D). These genes show differential regulation exclusively when both hormones (T_3_ and CORT) are present. Importantly, we found *dio2* to belong to the n_n_u cluster, meaning that its expression requires both T_3_ and CORT, whereas *dio3* expression only depends on T_3_ (cluster u_n_u).

These important results demonstrate that:-Most of CORT-regulated genes are also regulated by T_3_ in tailfin.-We provide a new list of CORT-only responsive genes, which are notoriously difficult to isolate [70].-Qualitatively, the effects of T_3_ and CORT co-treatment are not limited to the cumulative response from each hormone individually. There is a large palette of biological responses involving complex interactions between pathways (X-talks).-At the transcriptome level, transcriptional responses are dominated by the T_3_ effect and ADDITIVE, and although X-talks do exist, they correspond to ~10% of differentially regulated genes.

### 3.4. Understanding Molecular Phenotypes: From Lists of DE Genes to System Biology

To obtain an understanding of the processes involved for each type of biological response, we modeled a functional interaction network based on signaling and metabolic pathways available from the KEGG resources. Individual pathways summarize our current understanding of signal propagation and functional interactions between gene products in a given biological context (“Insulin signaling”, “Calcium signaling”, “Fructose and mannose metabolism”, etcetera), providing a well-focused and high quality knowledge base readily usable for biologists. An intrinsic property of biological pathways is that many factors are shared between pathways [71], and affecting the biological activity of one such factor can simultaneously affect multiple pathways. These factors often (but not always) correspond to hubs, which act as central communication points between network components. Therefore, pathways are not functionally independent from one another and fail (and are not aimed) to describe the overall complexity of functional interactions within the cell. Pathway analysis, as commonly carried out in RNA-Seq data analysis, cannot provide an integrated description of transcriptional remodeling. In contrast, a network of pathways clearly integrates the functional interactions between pathway components, and can help identify, for example, novel shortest routes between membrane receptors and transcription factors, thus corresponding to new signaling pathways [57]. To this end, all of the KEGG pathways containing at least one DE gene were collected and merged together, in a procedure similar to that of [56] (see Methods and Appendix A).

The reconstructed network is shown Figure 4A. It is built from 157 pathways and is composed of 3606 nodes (gene products) and 11,216 edges (functional interactions), in which 108 DE genes could be mapped: 50 from T_3_ response, 5 from CORT, 47 from ADDITIVE, and 6 from X-talks. Of note, 90% of X-talks genes interact with less than 13 other genes, while 90% of the ADDITIVE genes interact with up to 31 other genes, thus suggesting that ADDITIVE genes may engage more functional interactions within the network than X-talks. Unfortunately, this fails to reach statistical significance (*p*~0.1, permutation test, see methods), mostly because of the small number of DE genes mapped into the network. We then collected the first neighbors (i.e., genes DE and non-DE sharing a direct interaction) of each DE gene (Figure 4B). To our surprise, they all form strongly connected sub-networks, implying that they collectively participate in specific cellular functions. The corresponding biological processes are clearly in line with the expected phenotypes induced by treatments with a single hormone (Figure 4C): T_3_ effect is associated to developmental processes, signal transduction and metabolic processes, while CORT effect is associated to stress and immune responses. In contrast, ADDITIVE and X-talks effects differ markedly from the specific effect of each hormone alone (Figure 4C): mostly DNA damage, cell death, and metabolism for the former, and JAK-STAT signaling for the latter. This analysis shows that, contrary to what standard GO analysis would suggest (Figure 2D), the effects of the T_3_-CORT co-treatment are more complex and involve multiple components, each corresponding to a very specific biological response. Also, the enrichment of cell death terms in ADDITIVE genes after co-treatment is strongly indicative of a synergy between the two pathways.

If true, one would predict treatment with sub-optimal T_3_ concentration or CORT to have little effect on tail regression, but co-treatment should result in a strong acceleration. We took advantage of the fact that tail explants of pre-metamorphic tadpoles (NF54) can be cultured ex vivo for extensive periods of time (up to 7 days [72]) and subject them to various hormone treatments. As expected, treatment with nominal T_3_ concentration (10 nM) leads to rapid shrinkage and regression of the tissues (Figure 4D), with no apparent sign of necrosis nor tissue degradation. This recapitulates well the known fact that tail regression is an autonomous and local process [72,73]. When supplemented with 100 nM CORT, tail regression does not speed up, suggesting that at high T_3_ concentration, it already reached a maxima (not shown). In contrast, with sub-optimal T_3_ concentration (1 nM) or 100 nM CORT alone, explants are phenotypically un-affected, even after extensive incubation time. This indicates that each hormone, when acting alone at these concentrations, does not trigger tail regression. In contrast, co-treatment with sub-optimal T_3_ concentration and CORT strongly accelerate tail regression, without apparent necrosis. This is a clear confirmation of the functional synergy between CORT and T_3_ [42,49,52], and the implication of ADDITIVE genes in cell death pathways.

To complete the network analysis above and get more a mechanistic insight of each biological response, we reconstructed a PPI network. To this end, we collected all proteins known to physically interact with DE gene products from the BIOGRID database (see Methods). The resulting network is composed of 2481 nodes and 3277 edges, with a giant component followed be a number of smaller disconnected networks (Figure 4E). We used two metrics to characterize hubs: connectivity (a.k.a. degree) and betweenness centrality (Figure 4F) [74,75]. The latter, ranging from 0 to 1, measures how much the corresponding node influences the network, while the former measures how much the node is connected to other nodes of the network. The relationship between degree and betweenness centrality helps discard central nodes in star-like small networks (such as in Figure 4E, right) characterized by a low degree and a high betweenness centrality. Surprisingly, DE hubs are highly enriched in various genes involved in DNA methylation and belonging to the T_3_ (DNMT3 up, EZH2 up, MECP2 down, and UHRF1 down) or ADDITIVE (UHRF2 down) response types. This is a clear signal that tail regression might be mediated through changes in DNA methylation.

To further explore this relationship, we plotted a heatmap of the expression Log Ratio of 75 genes annotated in the GO terms related to DNA methylation. Results clearly demonstrate that most of them display increased or decreased expression relative to the non-treated control, and only a few genes display minute (or no) difference of expression (Figure 4G and Appendix A). With only a few exceptions, gene response is very similar when treatments are applied to tail culture or whole animals. Despite a few differences between treatments, T_3_ and T_3_-CORT profiles are very similar. The expression of 15 genes (*dnmt1*, *uhrf1*, *mecp2*, and *mbd2*, to name a few) is consistently decreased in response to all treatments. In contrast, *apobec2* expression is strongly induced in all experimental conditions, although it displays differences between culture vs. whole animal. Importantly, the expression of two key components of the DNA hydroxy-methylation machinery, *tet2* and *tet3*, is also consistently increased and displays clear signs of synergy. These results therefore strongly suggest that the DNA methylation machinery undergoes a strong transcriptional switch.

### 3.5. T_3_ and CORT Induced Massive and Complex Changes of DNA Methylation

To test whether the action of T_3_ and CORT on tail regression is mediated, at least in part, through changes of DNA methylation, we undertook to characterize the DNA methylation dynamics by MethylCap-Seq. Briefly, methylated DNA is captured by affinity column and released with buffers of increasing ionic strength. The methylated DNA captured is then deep sequenced to derive genome wide maps of DNA methylation changes. In this system, a peak corresponds to a region of increased or decreased DNA methylation. Visual examination of genome profiles quickly highlights a complex dynamic of DNA methylation. A few representative examples are shown Figure 5A–C. It is difficult to relate them to changes of gene expression, as they are located in gene deserts (Figure 5B) or nearby genes that do not respond to treatments (Figure 5A,C).

We further processed the data and identified regions of differential methylation levels (DMR). We found a non-redundant set of 17,705 DMRs, with extensive numbers in all treatment conditions: 7000, 6358, 8324 after T_3_, CORT, and T_3_-CORT treatments, respectively (Appendix A). The dataset is dominated by demethylation (94.3% to 97%), despite a low but non-negligeable fraction of re-methylation (3.0% to 5.7%). Most DMRs (90%) are the genomic region up to 1.1 kb large, and there is no shift in DMR size upon treatment (Figure 5D). We also found that the ratio of CpG over other di-nucleotides is higher in DMRs when compared to exonic sequences (Figure 5E), indicating that DMRs are protected from spontaneous deamination of 5 methyl-cytosines.

Altogether, these results unambiguously show that each treatment alone or in combination induces complex and extensive changes of DNA methylation.

### 3.6. The Cross-Talks-Like Complex Regulations Drive the Majority of DNA Methylation Changes

We next clusterized DMRs by using the same procedure as for RNA-Seq, and we derived response types corresponding to T_3_, CORT, ADDITIVE, and X-talks effects (Figure 6). The naming convention based on a ‘d’, ‘u’, and ‘n’ triad is also similar. Qualitatively, DMRs follow an inverse dynamic of transcriptome (Figure 6A–D), where X-talks correspond to the vast majority of DMR responses (65.4%, 11,576/17,705), while T_3_ and CORT effects are only a minority of responses (7.1% -1263/17705- and 6.7% -1,199/17,705-, respectively). ADDITIVE responses account for 20.7% of DMRs (3667/17,705). X-talks DMRs are actually composed of a large diversity of responses, which is indicative of multiple regulatory mechanisms (Figure 6D). First, the action of one hormone inhibits the effect of another, as with clusters d_n_n (2633 DMRs) and u_n_n (41 DMRs), where CORT cancels the effect of T_3_, and inversely for the clusters n_d_n (185 DMRs) and n_u_n (235 DMRs), where T_3_ cancels the effect of CORT. Another category (14.1%, 1632/11,576) corresponds to opposite responses with one or two hormones, as displayed for clusters d_u_u, d_d_u, u_n_d, u_u_d. Intriguingly, for 2064 DMRs (17.8%), CORT alone has no effect on the DMRs methylation state but co-treatment ends up with opposite regulation of the T_3_ treatment alone (d_n_u cluster). Lastly, two clusters, n_n_u (378) and n_n_d (94 DMRs), only display variations of methylation level when both hormones are present and none otherwise. Overall, these results demonstrate that (1) T_3_ and CORT treatments induce a large palette of DNA methylation changes, and (2) transcriptomic and DNA methylation data follow opposite trends, where X-talks correspond to a minority of biological responses in one case, and a majority of responses in the other.

### 3.7. DMRs Are Located Far from Genes

To get a better understanding of DMRs distribution, we compared their genomic location to that of genes. We found that 27.7% of DMRs (4913/17,705) overlap with, or are located within, genes (Figure 7A). This is in line with DNA methylation changes reported in the brain of metamorphosing tadpoles [76]. We note, however, that in our case, the repertoire of DE genes is very limited and corresponds to only five genes: *B4GALNT4* (d_d_d), *CHTF18* (d_n_d), *PAPPA* (n_n_u), *ATP12A* (n_u_n), and *ANGPTL2* (u_n_u). Other DMRs are located far away from genes, typically from 10 to 100 kb away. All four types of biological response provide equivalent genomic distribution. This result is not surprising, as enhancers and response elements are expected to be located at large distances from their target gene, and even enhancers located within a gene may in fact regulate a different target [21,58]. We next addressed whether the presence of transposable elements or other repeated sequences correlate with biological responses. We found that most DMRs, from 60 to 75%, overlap with known transposable elements and repeated sequences (Figure 7B), and they all share similar proportions of individual repeated sequence families (Figure 7C). We did not find any feature shared by DMRs that do not overlap with TEs, with the notable exception of the PTR_XL family.

We next addressed whether DMRs are enriched in predicted binding sites for nuclear receptors (NRBS), which mediate the nuclear action of CORT and T_3_. NRBS are composed of two AGGTCA half sites in various orientations relative to one another, and spaced by a linker of varying sizes, typically 0 to 8. The glucocorticoid receptor (GR) is preferentially found at inverted repeats with a 3 bp spacer (IR3) [77], whereas thyroid hormone receptor (THR) binding is biased toward direct repeats with a 4 bp spacer (DR4) and sometimes 6 bp (ER6) [78,79]. NRBS are composed of two 6 bp motifs in various orientations and spaced by a linker of varying sizes, typically 0 to 8. As expected [66,80], known DNA binding motifs of THR (DR4, ER6, and IR0) correspond to the vast majority of NRBS found in DMRs (Figure 7D). Only a limited fraction of IR3 motifs (GR binding sequences) could be found in DMRs and the genome. This certainly reflects over-fitting of the model as a result of poor initial training of IR3 motifs [67]. Overall, motifs found in DMRs are slightly enriched in DR4 compared to other motifs (Chi square test, *p* = 0.049), whereas ER6 are somewhat under-represented (Chi square test, *p* = 0.049).

## 4. Discussion

In this work, we interrogate the functional interactions between TH and GC signaling pathways. Our model is Xenopus metamorphosis, a well-known developmental transition dependent on both TH and GC signaling. In addition to their developmental role, GC also mediate stress response, resulting in a set of complex interactions.

GC potentiate the action of TH by modulating its synthesis or the balance between the metabolic activation of the hormone from a (less active) precursor or its degradation. On the other hand, in mammals, TH also exert a central regulation on GC synthesis [81,82]. This model of TH and GC signaling interactions would predict only a reciprocal modulation of the effect of each hormone individually, not complex types of biological responses at the gene level; this is the point that we are challenging.

### 4.1. TH and GC X-Talks: A Large Repertoire of Transcriptional Regulations

Anuran metamorphosis is a fast and well-choreographed process. Limb growth and development is a very early process preceding tail resorption by several NF stages, therefore ensuring proper locomotion throughout the entire process [53]. The timing of tail resorption is controlled by modulating the level of available TH with DIO2 and DIO3, which expression is in turn modulated by GC [42,83]. This model implies that tail resorption is fundamentally controlled by TH and does not predict X-talks response genes. Our data clearly demonstrate that this model is incomplete and that T_3_ and CORT act on a specific set of genes.

The first indication that gene-level regulation is modulated by TH and GC may be more complex was carried out by Kulkarni and Buchholz [52], where they treated Xenopus tadpoles with relatively high doses of T_3_ (50 nM) and CORT (100 nM) and examined gene expression in tails with microarrays. They found a large set of genes (~5 k) with altered expression levels in response to one or both hormones. Among these, hundreds of genes (1308) presented an atypical expression profile, where they were exclusively over-expressed (or repressed) when both hormones are present. Unfortunately, these analyses did not benefit from the powerful combination of clustering and system-biology level modeling, thereby preventing deeper analysis.

Therefore, we undertook to address it by deriving a detailed map of the changes of functional state throughout the genome. At the molecular level, we measured gene expression changes in response to hormone treatments, and we modeled RNA-Seq signals into system-wide integrated biological responses. Combining the sharp and contrasted phenotype of tail regression together with the exquisite sensitivity of functional genomics and network analysis, we demonstrate that the transcriptional response to TH and GC of pre-metamorphic tadpoles is more complex than initially envisioned. The key result is that despite the fact that changes of gene expression are dominated by the effect of each hormone individually (T_3_, CORT, or ADDITIVE), an extra set of complex regulations sum up to a significant fraction of transcriptional responses. This can be summarized in the following equation-like format: TH **∙** GC > TH + GC, meaning that the joint effect of both hormones is more than/not limited to the sum of effect of each hormone individually.

X-talks profiles are not trivial and cannot be accounted for with additive effects. For instance, the u_n_d profile (Figure 3D) implies that, despite an apparent lack of action, CORT inverts the regulation. For others, the action of one hormone dominates over the other (e.g., d_u_d and u_d_u) or they cancel each other (e.g., d_n_n and n_u_n). All these cases unambiguously correspond to regulatory interactions between TH and GC signaling in individual genes, and not the simple addition of their independent effect on target gene expression.

The two clusters n_n_d and n_n_u are important: their expression is strictly dependent on the action of both hormones simultaneously and using single hormone treatments will fail to identifying them. The corresponding gene products are involved in membrane bound signaling, or correspond to extracellular proteases and components of the extracellular cellular matrix. It is noteworthy that the transmembrane TH transporter MCT8 belongs to the n_n_u cluster; this suggests that part of the tailfin resorption program is strictly dependent on the simultaneous action of both hormones. Buchholz previously demonstrated that GR knock out mutants die at metamorphic climax, at the time of tail regression [46]. Strikingly, the phenotype of *pomc* mutants could be rescued with the addition of CORT but also with high doses of T_3_, suggesting that the function of CORT would only be limited to the potentiation of TH action. Our results suggest that alternative and partly redundant pathways might be involved, instead. In line with this, the expression of the MCT10 transporter, which is functionally redundant with MCT8, is regulated by T_3_ but not CORT (u_n_u cluster).

The gene regulatory networks controlling transcription can be decomposed into a number of specific genetic sub-circuits implementing various regulatory logics [84]. The transcriptional output can be viewed as the end result of the processing from several sub-circuits integrating regulatory input signals with combinatory logic. Intuitively, X-talks profiles can also be described with boolean operators obeying some regulatory logic. More formally, the n_n_u and n_n_d responses can be rephrased with an AND boolean operators where change of gene expression (up or down) depends on the regulatory logic T_3_ AND CORT. The same is true with n_u_n, but with the exclusive OR operator (T_3_ XOR CORT), and for u_n_n with the NOT operator (NOT (T_3_ OR CORT)). More complex responses, especially when signal is inverted (e.g., u -> d) can be further build from a combination of additional logic blocks: d_d_u would result from INV (T_3_ OR CORT). The point here is that beyond a clear and explicit representation, the simple fact that the different X-talks responses obey different regulatory logic imply that in each case, gene expression is governed by a very specific regulatory circuit(s). Even though the molecular mechanisms of each type of transcriptional response needs to be precisely dissected, there is little doubt that a large number of specific regulatory mechanisms are expected. This clearly extends the initial model much further, where TH and GC pathways would regulate hormone availability.

Our data fit well with the known biology of THR and GR. Regulation of gene expression by THR is well described with a dual model, where the receptor acts as both a repressor and an activator of transcription by the differential recruitment of transcriptional co-repressors and co-activators [85]. The THR can also display very gene specific regulations [69,79]. GR action involves a remarkably large number of molecular pathways which can clearly implement various regulatory logics.

### 4.2. X-Talks: An Unexpected Large Impact on DNA Methylation

We found that TH and GC signaling cooperate to alter the transcript levels of a majority of the factors involved in DNA methylation and demethylation pathways. Monitoring genome wide DNA methylation changes by MethylCal-Seq further provide strong evidence of massive and large scale demethylation, although some local increases of DNA methylation also occur. Contrary to our expectations, these changes in DNA in methylation follow complex regulations and are mostly dominated by X-talks with little T_3_, CORT, and ADDITIVE responses. This important result, which contrasts sharply with the transcriptome output dominated by T_3_ and ADDITIVE responses, implies that distinct regulatory processes occur at each level and that changes of DNA methylation are more dynamic and more dependent on hormonal environment inputs than initially thought. Clearly, the extent of genomic DNA methylation changes is buffered and does not translate directly into changes of the transcriptome.

In xenopus tadpole brain, *klf9* transcription is induced upon treatment with T_3_ or CORT, but much stronger with T_3_-CORT [86,87]. At the mechanistic level, *klf9* transcriptional expression is sustained by the binding of THR and/or GR transcription factors at a short genomic region located a few kbs upstream, and containing an enhancer called *klf9* Synergy Module (KSM). The progressive DNA demethylation at the KSM correlates with accumulation of TET3 and the progressive transcriptional induction of the gene [88]. Note that this ‘synergy’ of action fits with the ADDITIVE response in the terminology we use in this work. Interestingly, DNA methylation levels at KSM do not seem to be affected T_3_ nor CORT in tailfin despite a clear ADDITIVE transcriptional induction. Therefore, the DNA methylation dynamic relies (at least in part) on tissue specific factors.

Nonetheless, our data are well in line with previous reports demonstrating a strong DNA demethylation, together with local DNA re-methylation, in gene bodies during metamorphosis in brain [76]. We also found DMRs located not only at gene coding sequences, since the majority of them are located in non-coding regions of the genome (intergenic). This apparent discrepancy likely reflects differences in the processing pipe-lines and the fact that we explored non-coding sequences. It is often difficult to interpret the data because these regions are packed with repeated sequences and are poorly annotated, therefore preventing a direct comparison with functional elements. In line with this, most DMRs could not be related to other functional changes or gene features. Nonetheless, we found a clear enrichment of a specific transposable element family (PTR_XL) in X-talks and ADDITIVE DMR clusters. Although we could not detect any obvious enrichment of predicted nuclear receptor binding sites in these sequences, the demethylation of DNA at these loci strongly suggests they are being transcribed, which is reminiscent of previous work where SINE-like TEs were found to be strongly expressed during metamorphosis in brain [89].

In brain, immuno-reactivity for 5-hmC, 5-caC, and TET3 increase in parallel in the brain of tadpoles undoing spontaneous metamorphosis [49,76], and particularly at the metamorphic climax where TH levels are highest [90]. Their correlation indicates, at least in part, that TET3 might be involved in the demethylation process. MethylCap-Seq analysis in tadpole brain also displayed DNA demethylation, suggesting that this would be a T_3_ dependent mechanism. Raj et al., [88] provided additional support by treating animals with T_3_ and displaying accumulation of TET2-3 and biochemical DNA demethylation intermediates (5-hmC, 5-caC), strengthening their point. Our work extends from this as we demonstrate that the vast majority of DNA methylation changes are not only dependent on T_3_, but also depend on CORT. This implies that the regulatory processes involved are more complex than initially thought.

What could the regulatory mechanisms involved in the genome wide change of DNA methylation be? Many components of the DNA methylation and demethylation machineries functionally interact with TH signaling. The *dnmt3a* gene is a TH direct target gene [91,92] (but its expression is not regulated by CORT, our data), while TET3 physically interacts with THRa [93]. TH induces TET2 expression [88] and regulates the expression of *gadd45a* and *b* [94,95], *mbd3* [96], and *apobec* [97]. GC signaling, too, is functionally connected to DNA methylation/demethylation [98]. In a more indirect manner, GR also physically interacts with the histone methylase TRIM28 [99] and a component of the chromatin remodeling complex SMARCA4 [100], which in turn impact DNA metylation. GR is a very versatile platform that can accommodate numerous action mechanisms [101,102].

The fact that all but one DE hub acting on DNA methylation display T_3_ responses may seem counter intuitive when DMR dynamics display mostly X-talks properties; several lines of evidence can help settle this apparent discrepancy. First, our experiments measure changes of gene expression and many events may be regulated at other levels (post-translational modifications, alterations of intracellular Ca++ signaling, etcetera). Second, even though UHRF2 is the only non-T_3_ hub (ADDITIVE) of the PPI network, it nonetheless makes a functional connection between DNA methylation and transcriptional regulation by T_3_ and CORT. Furthermore, the functional response is certainly not limited to hubs, and other non-hub factors may be involved. Two interesting candidates are OTUD4 and ZMPSTE24, for which the transcriptional response is also ADDITIVE (d_d_d and u_u_u, respectively). A quick BIOGRID survey indicates that in humans, UHRF2 and OTUD4 engage in physical interactions with USP7 (expressed in tailfin, but not DE), thus providing a good starting point to elucidate the molecular cascades involved.

## 5. Conclusions

In this work, we demonstrate that TH and GC pathways cannot be considered in isolation. Rather, they intimately cooperate and produce complex regulatory interactions, both in terms of gene expression and DNA methylation landscape. The sheer level of X-talks changes of DNA methylation is not reflected at the transcriptome level, and was fully unexpected. This has important implications for all those working in clinical and non-clinical environments manipulating these two signaling pathways.

## Figures and Tables

**Figure 1 cells-10-02375-f001:**
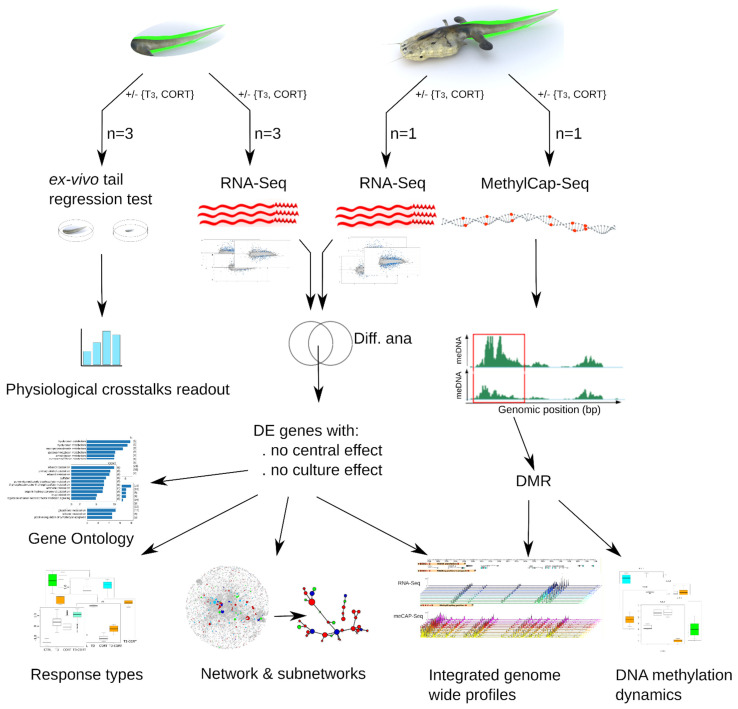
Experimental setup to address T_3_ and CORT cross-talks. An experimental and data analysis workflow. Tailfin (green) response to treatments was characterized by functional genomics. For each experiment, samples were collected from pools of five tailfins.

**Figure 2 cells-10-02375-f002:**
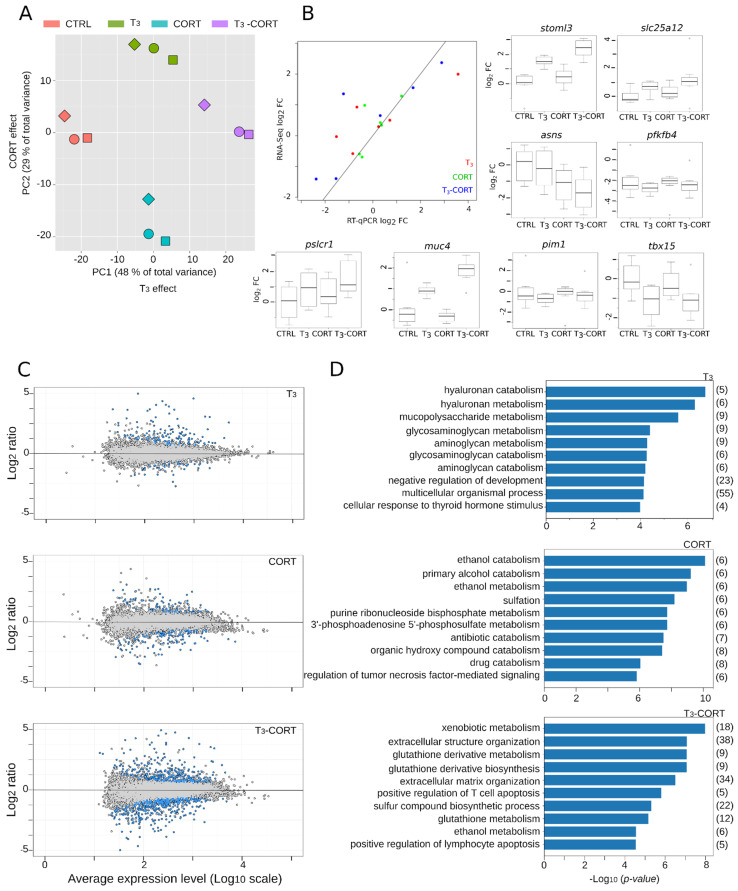
Transcriptome analysis following T_3_ and/or CORT. (**A**) Principal Component Analysis (PCA) of RNA-Seq variance from tailfin dissected from cultured tail explants. The two main components capture the effects of both hormones, corresponding to 77% of the total variance. Square, circle, and diamond: biological replicates. (**B**) Independent validation of RNA-Seq data by RT-qPCR. Scatter plot of the relationship between RNA-Seq versus RT-qPCR expression changes, also shown in box plots. (**C**) Average expression value versus fold change (MA plot). DE genes are in blue. (**D**) Gene Ontology analysis.

**Figure 3 cells-10-02375-f003:**
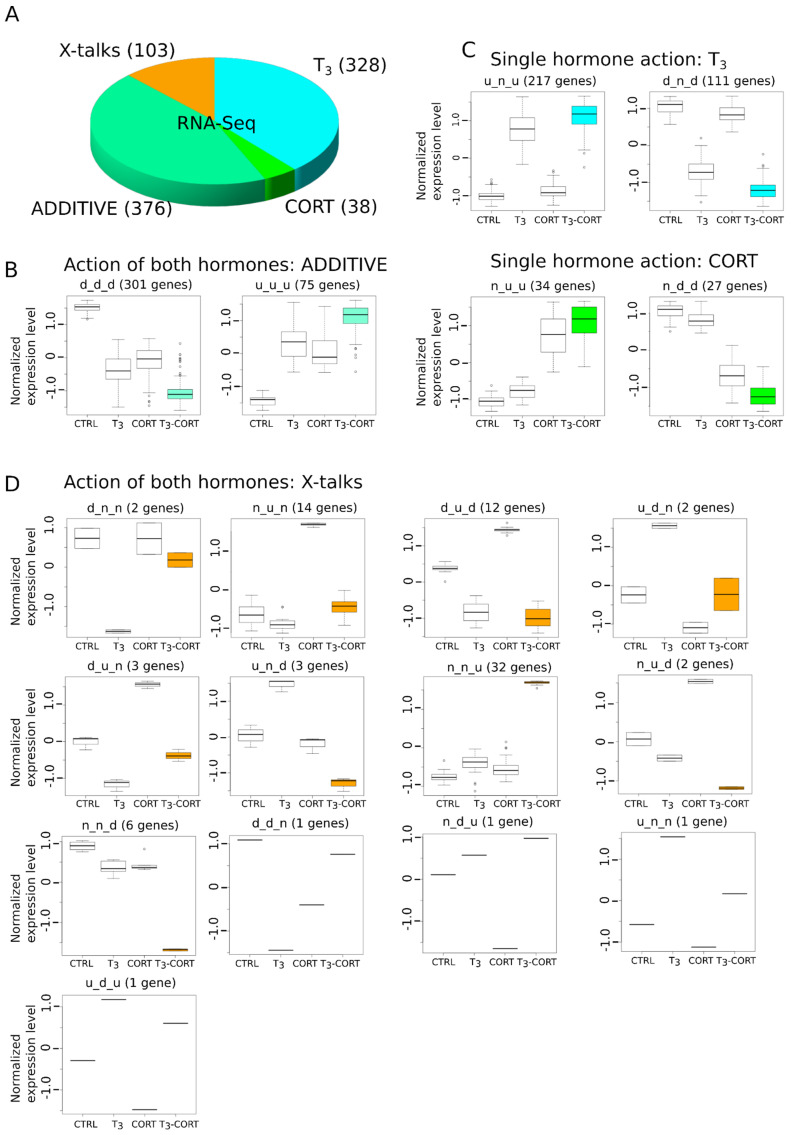
Gene expression is modulated by one, the other, or both hormones. Only genes displaying consistent expression between explants culture and whole animals were kept. (**A**) Types of gene regulation by T_3_ and CORT. (**B**) Gene regulation by either T_3_ or CORT. (**C**) Gene regulation by T_3_ and CORT. (**D**) Complex regulation by both hormones (X-talks).

**Figure 4 cells-10-02375-f004:**
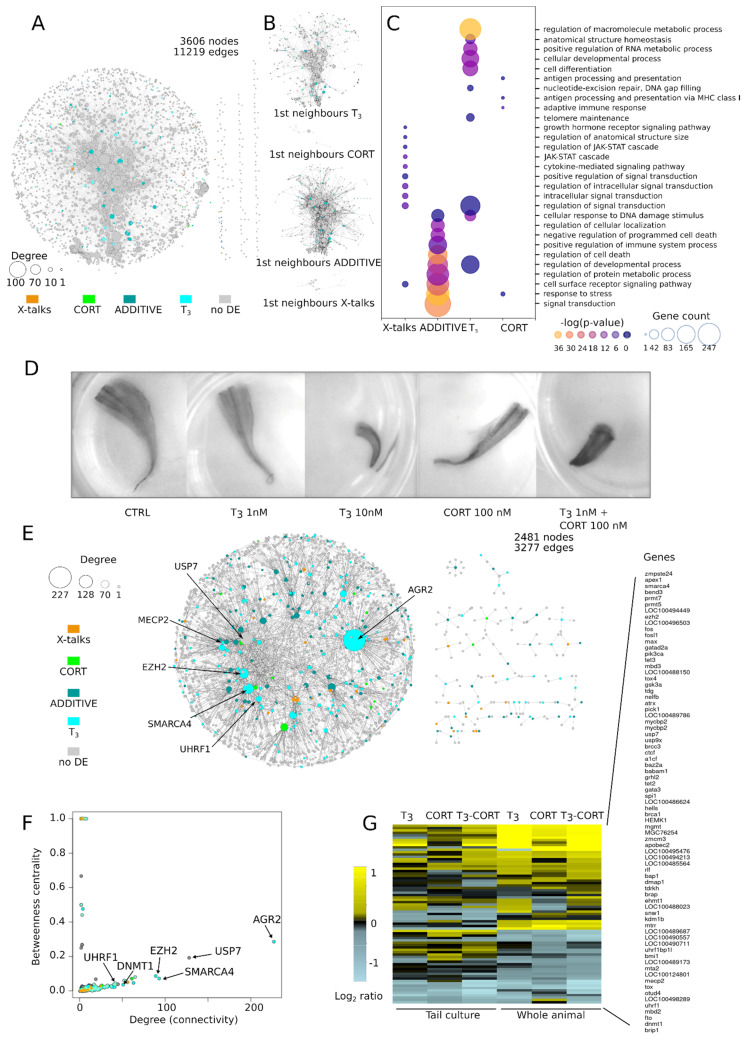
System level modeling of transcriptional responses identify the DNA methylation dynamic as a major mediator of tailfin regression. Only transcriptional responses displaying consistent expression between explants culture and whole animals are considered. (**A**) Network of KEGG pathways. (**B**) DE genes together with their first neighbors in the network, forming densely connected sub-networks. (**C**) GO analysis of the sub-networks highlighting terms such as cell death, DNA damages, and DNA repair. (**D**) Tail regression test in vitro. Synergistic action of T_3_ and CORT after 3 day treatments. (**E**,**F**) Protein-Protein Interactions network and identification of hubs. Most DE protein complexes relate to DNA methylation. (**G**) Heatmap of expression levels of genes involved in DNA methylation.

**Figure 5 cells-10-02375-f005:**
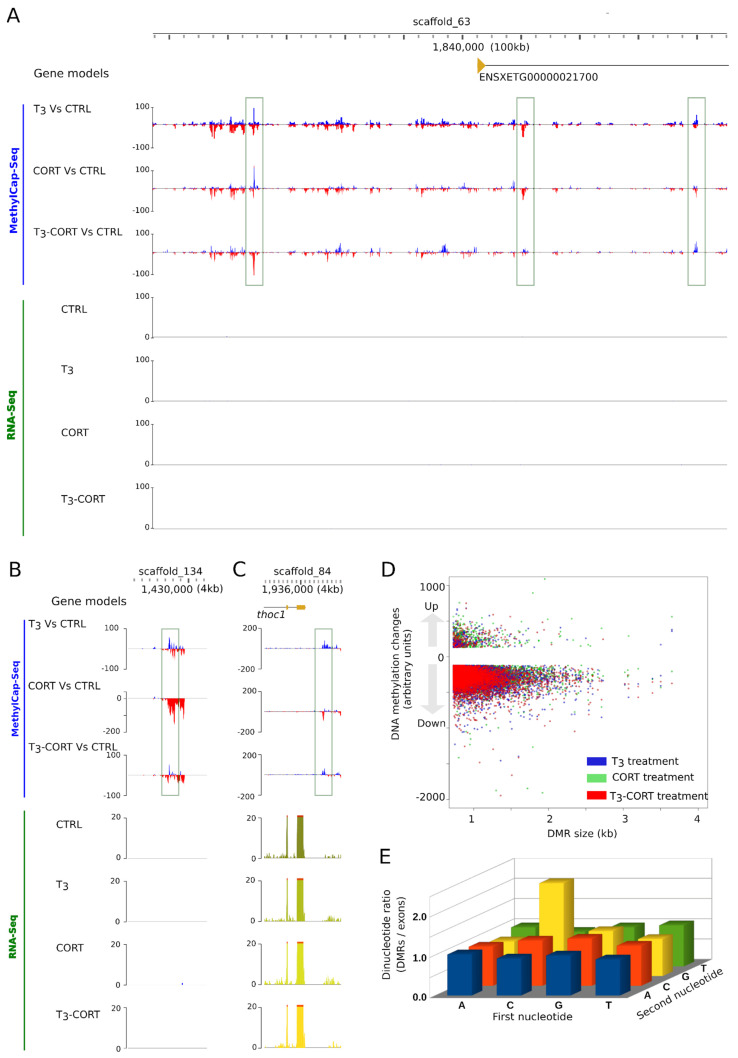
T_3_ and CORT induce genome wide changes of DNA methylation levels. (**A**–**C**) Three independent loci with local changes of DNA methylation (green arrow). Tracks order: gene annotation, DNA methylation changes relative to CTRL (T_3_, CORT, and T_3_-CORT), mRNA abundance (CTRL, T_3_, CORT, and T_3_-CORT). (**D**) Relationship between the genomic span and the amplitude of differentially methylated regions (DMR). (**E**) Differential dinucleotide frequency found in DMRs and exonic sequences. DMRs are enriched in CpG.

**Figure 6 cells-10-02375-f006:**
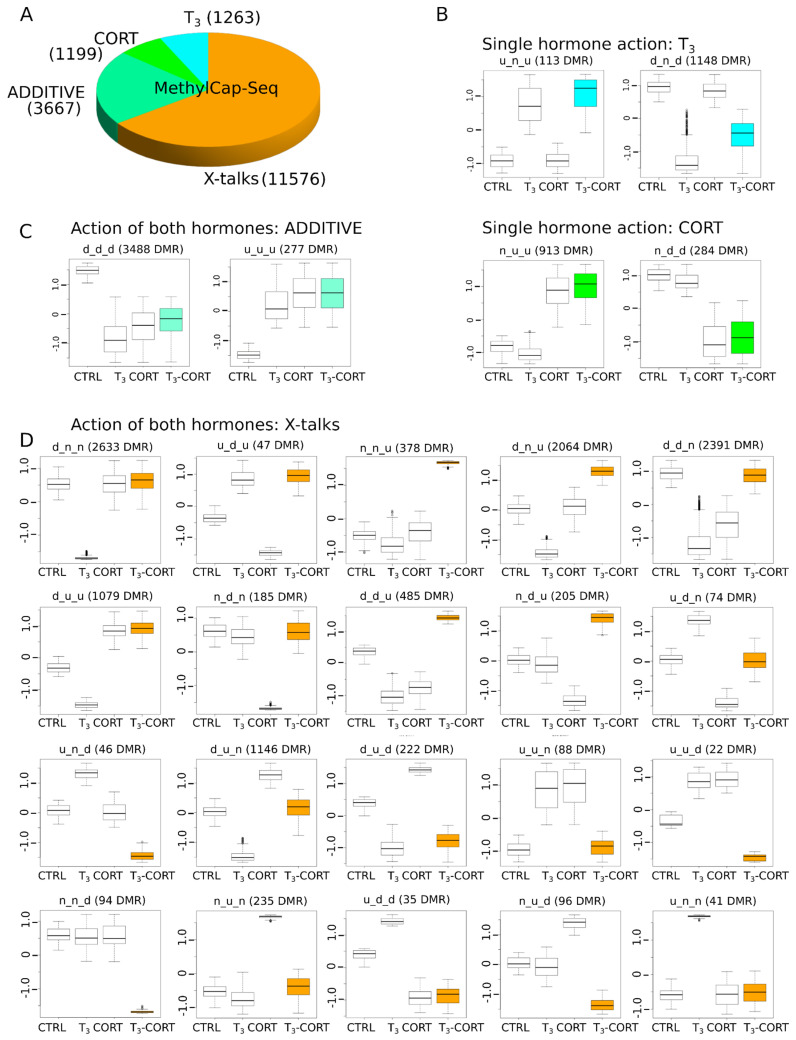
The complex dynamics of DNA methylation levels at DMRs. (**A**) Types of gene regulation by T_3_ and CORT. (**B**) Gene regulation by either T_3_ or CORT. (**C**) Gene regulation by T_3_ and CORT. (**D**) Complex regulation by both hormones (X-talks).

**Figure 7 cells-10-02375-f007:**
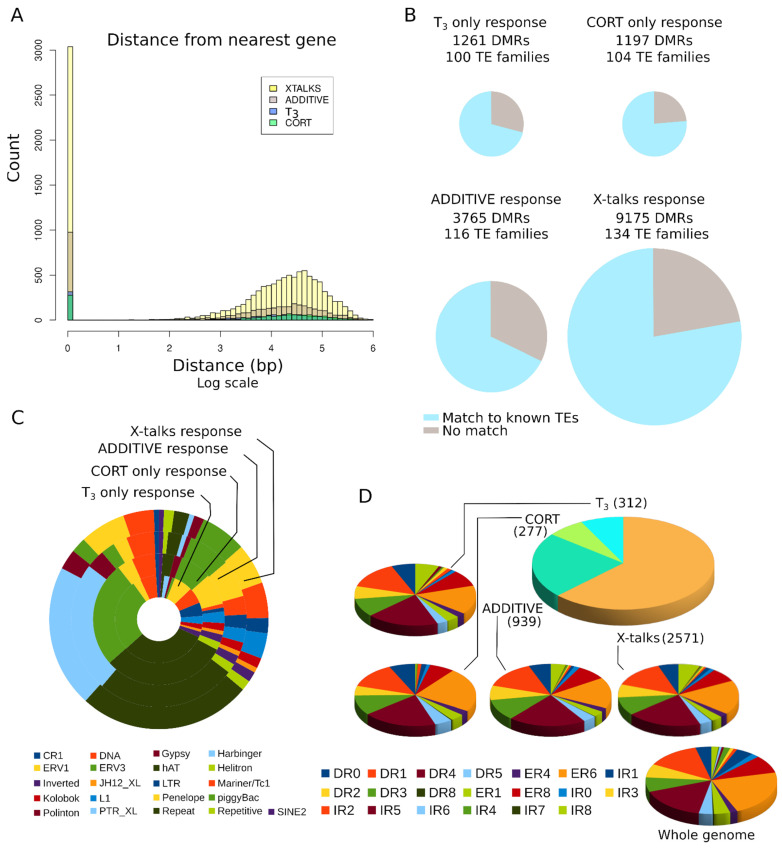
DMRs are located close to and far away from genes. (**A**) Distance between DMRs and the nearest gene. (**B**) Overlap between DMRs and known transposable element (TE) sequences. (**C**) TE content of DMRs. (**D**) Nuclear receptor binding sites predicted in DMRs. Only DMR containing predicted NRBS are shown.

## Data Availability

Raw reads were deposited at SRA under the reference PRJNA748587.

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
