# Peer review of "Transcriptome and Methylome Analysis Reveal Complex Cross-Talks between Thyroid Hormone and Glucocorticoid Signaling at Xenopus Metamorphosis"

_cells, 2021, doi:10.3390/cells10092375_

Round 1
Reviewer 1 Report
The authors analyze the tadpole tailfin by RNA-sequencing and DNA methylation, with the goal of determining the relative response to thyroid hormone (T3) and glucocorticoid (GC). Using computational analyses, the authors report complex interactions in the response to both hormones, which they broadly discuss as "cross-talks" or "x-talks". Although these interactions have been relatively little studied previously the results emphasize the importance of combined hormonal signaling in gene expression and development. These results are interesting.
The main comments concern simplifying and clarifying the text with respect to the take home message for a non-specialist reader.
Comments:
- The term "cross-talks" is used frequently in the text and figures, but the term is vague. It would help to define more explicitly what type of x-talk is intended, if possible. Is it possible to identify one or two major forms of x-talk and discuss these in more specific terms? For example, why not use more commonly understood terms like "synergy" or "cooperation"?
- Some text sections are difficult to follow, such as p. 10, top paragraph ("Overall, (Fig 3A), a majority of genes..."), p. 16 top paragraph ("The naming convention based on a d, u, n, triad..."). The dense use of abbreviations "d, u, n" is not easy to read and the meaning may be clearer, for example, by breaking up the paragraph or using simpler terms that are more widely understood.
- Technical.
3a. Please explain number of replicates in the flow diagram (Fig 1). How many tailfins? Are these pools or single tailfins? The information is not easy to locate here or in Methods.
3b. Please explain the time duration of treatments (Fig 1).
3c. Was MethylCap-Seq performed on a different date for each replica?
Was it possible to confirm the data of MethylCap-Seq using ex-vivo tail?
- Figure 4G, gene name list should be enlarged - font is too small to read.
- Typos - please check throughout. A couple of examples:
- 10, Line 8 down, ADDIVIVE
- 19, GC potentiates the action...
- 2, GR proves to be a very versatile...
- reference corrections. Check the entire reference list.
Several citations require author names (listed now only by initials, e.g. Buchholz, DR & DR, B, 2015).
Also, in the text. One example, p. 6, line 3 down, LB et al, 2015, RN et al, 2005.
Author Response
First, we would like to thank the referee for providing critical comments on the manuscript and thereby helping us improving it. Please, find below our response to individual comments.
Comments
1) The term "cross-talks" is used frequently in the text and figures, but the term is vague. It would help to define more explicitly what type of x-talk is intended, if possible. Is it possible to identify one or two major forms of x-talk and discuss these in more specific terms? For example, why not use more commonly understood terms like "synergy" or "cooperation"?
=> The point of our article is to describe the repertoire of possible interactions between signaling pathways by measuring changes of gene expression. Of course, this includes already known response types such as synergy or cooperation, but this is not limited to them. We therefore choose a more general term, that can encompass other yet-unknown responses. We define the terms cross-talks and X-talks as such, page 5: "From herein, we will refer to the complete set of functional interactions between pathways with the terms "cross-talks" or "X-talks", and this collectively accounts for already known mechanisms of action (synergy, cooperation, antagonism) and well as any novel mechanism of action".
For clarity, references to 'cross-talks' before we introduced the term were reworded. This is found at the following locations in the manuscript:
- Page 4 "Nuclear receptors have a natural tendency to cross-talk with each other, thus making their corresponding pathways cross-talk too", was replaced by "Nuclear receptors have a natural tendency to functionally interact with each other, thus making their corresponding pathways cross-dependent of one another"
- Page 4 "Despite numerous evidences that TH and GC pathways functionally interact, very little details are known about the mechanisms involved and the general properties of these cross-talks" replaced with "Despite numerous evidences that TH and GC pathways functionally interact, very little details are known about the mechanisms involved and their general properties".
- Page 5 " Xenopus metamorphosis is an outstanding model to describe the diversity of cross-talks", replaced with " Xenopus metamorphosis is an outstanding model to describe the diversity of functional interactions between pathways"
2) Some text sections are difficult to follow, such as p. 10, top paragraph ("Overall, (Fig 3A), a majority of genes..."), p. 16 top paragraph ("The naming convention based on a d, u, n, triad..."). The dense use of abbreviations "d, u, n" is not easy to read and the meaning may be clearer, for example, by breaking up the paragraph or using simpler terms that are more widely understood.
=> We rephrased the corresponding sections.
Concerning the nomenclature based on the d, u, n triad, we understand that it is very formal, but we feel it has the benefit of being very explicit and dissipate the ambiguities introduced by natural language. This is not a trivial issue, that we encountered many times when discussing with colleagues. To help the reader, we extended the description of this compact notation in the 'clustering' section of METHODS. The first section now reads:
Clustering of DE genes is aimed at classify individual genes into a number of a specific "response type". Expression values of each gene across the four treatment conditions (CTRL, T3, CORT, T3-CORT) were standardized by setting their average to 0 and their variance to 1. For each treatment, the normalized gene expression level is compared to CTRL and used to derive whether it is up- ('u'), down- ('d') or not- ('n') regulated after each treatment. Genes are then assigned to a cluster named after the corresponding letters arranged in the T3, CORT, T3-CORT order. This compact notation nicely summarized transcriptional responses. For example, gene transcription only induced with T3 is labeled u_n_u: transcription is up with T3 (first 'u'), not affected with CORT (middle 'n'), and up after T3-CORT co-treatment (last 'u'). Similarly, CORT only responsive genes are n_u_u or n_d_d, and genes transcription regulated by both T3 and CORT belong to d_d_d or u_u_u. Non trivial regulations (i.e. X-talks) also become explicit. For instance, u_n_n corresponds to a transcription level up in T3, but with no change after CORT and T3-CORT treatments.In this case, despite no direct action on its own, CORT cancels the action of T3.
Technical.
3a. Please explain number of replicates in the flow diagram (Fig 1). How many tailfins? Are these pools or single tailfins? The information is not easy to locate here or in Methods.
=> We updated Fig 1 accordingly. Pools of 5 tailfins were used for each experiments. We also updated the Fig 1 legend, by adding "Tailfin (green) response to treatments was characterized by functional genomics. For each experiment, samples collected from pools of five tailfins"
3b. Please explain the time duration of treatments (Fig 1).
=> Fixed.
3c. Was MethylCap-Seq performed on a different date for each replica?
=> Yes. They really are true, independent biological replicates.
Was it possible to confirm the data of MethylCap-Seq using ex-vivo tail?
=> No. We did not carry out the MethylCap-Seq on tailfin cultured ex-vivo.
Figure 4G, gene name list should be enlarged - font is too small to read.
=> Fixed.
Typos - please check throughout. A couple of examples:
10, Line 8 down, ADDIVIVE
19, GC potentiates the action...
2, GR proves to be a very versatile...
reference corrections. Check the entire reference list.
Several citations require author names (listed now only by initials, e.g. Buchholz, DR & DR, B, 2015).
Also, in the text. One example, p. 6, line 3 down, LB et al, 2015, RN et al, 2005.
Reviewer 2 Report
TH is necessary and sufficient to initiate the amphibian metamorphosis, while CORT acts synergistically with TH to accelerate progression of TH-induced metamorphosis and is essential for survival at its climax, strongly suggesting the cross-talks between TH and CORT. The authors conducted transcriptome analysis using highly TH responsive tailfin tissues treated with TH and/or CORT, constructed protein-protein interactions network, and found out that DNA methylation system is related to the most protein complexes that were differentially expressed after hormone treatment. They showed by MethylCap-Seq that more than 60% of differentially methylated regions doesn't display T3-only response, CORT-only response nor ADDITIVE response, but belongs to X-talks DMR (the other DMRs) in tadpole tailfin, whereas only 12% of TH and/or CORT-responsive genes are X-talks genes.
This manuscript should be accepted after minor revision.
Introduction
1) Too long introduction makes the roles of TH and CORT in amphibian metamorphosis obscure and difficult to understand. The authors should shorten it and clearly indicate "TH is necessary and sufficient to initiate the amphibian metamorphosis, while CORT acts synergistically with TH to accelerate progression of TH-induced metamorphosis and is essential for survival at the climax."
2) p. 3, line 8-15 and 20-21
"the known action of GC on dio2/dio3 can not explain the diversity of transcriptional responses in tailfin."
How are dio2/dio3 genes regulated after TH and/or CORT treatment in your experiment?
Results & Discussion
1) The authors should give the information of material (cultured tail explants or tailfins of tadpoles) in legends of Figs 2-4, and incubation time in legend of Fig. 4, especially about D.
2) Letters are too small in size to recognize in Figs. 2-6.
3) The authors should explain the principal component analysis in Fig. 2A and the meaning of the character positions (T3, CORT, and T3-CORT) more in detail.
4) The total number of DE genes is 729 in page 8, but this is 845 in page 10 (Fig. 3A). Why?
5) Almost all DE hubs involved in DNA methylation belong to T3 response type (Fig.4E), but 65.4% of DMRs are X-talks response type (Fig. 6A). Why?
6) 27.7% of DMRs overlap with, or are located within genes (Fig. 7A). Do these 3000 genes contain any genes involved in tailfin regression? If it is true, what are they? Which response type do they display?
7) p.14, line 15-18
"Results clearly show that most of them display increased or decrease expression relative to the non- treated control, and only a few genes display minute (or no) difference of expression (Fig 4G). "
"With only a few exceptions, gene response is very similar when treatments are applied to tail culture or whole animals. "
I cannot recognize them. Show the quantitative data to demonstrate them.
8) There are many error candidates.
p.10, line 8
ADDIVIVE--->ADDITIVE
- 11, legend of Fig. 2
- Gene regulation by either T3 or CORT. C. Gene regulation by T3 and CORT.
-----> B. Gene regulation by both T3 and CORT. C. Gene regulation by either T3 or CORT.
p.14, line1-2
, with a giant component followed be a number of smaller disconnected networks
-----> , with a giant component followed by a number of smaller disconnected networks
p.17, line 7
located within, genes (Fig 7A). ----> located within genes (Fig 7A).
p.14, line 15
Results clearly show that most of them display increased or decrease expression ----> Results clearly show that most of them display increased or decreased expression
- 14, line 40
(Supplementary Table S5) ----> Where is Supplementary Table S5?
- 15, legend of Fig. 5
(green arrow) ----> (green box)
p.16, line 2
(Fig 6A-C) ----> (Fig 6A-D)
- 16, line 6
, which is indicative a multiple regulatory mechanisms ----> , which is indicative of a multiple regulatory mechanisms
- 19, line 34
we show that tailfin of pre-metamorphic tadpoles, the transcriptional response to TH and GC of is more complex than initially envisioned. ------>
we show that the transcriptional response to TH and GC of tailfin of pre-metamorphic tadpoles is more complex than initially envisioned.
Author Response
We would like to thank the referee for providing an in-depth review of our manuscript, and by raising points that altogether helped us improving it.
Introduction
1) Too long introduction makes the roles of TH and CORT in amphibian metamorphosis obscure and difficult to understand. The authors should shorten it and clearly indicate "TH is necessary and sufficient to initiate the amphibian metamorphosis, while CORT acts synergistically with TH to accelerate progression of TH-induced metamorphosis and is essential for survival at the climax."
=> We reworked the introduction, and added the statement. Some part of the text, describing the structure of the different nuclear receptors binding sites, has been moved to the corresponding results section (DMRs are located far from genes). We agree that this change and the addition of makes the introduction more focus and straight tot the point.
2) p. 3, line 8-15 and 20-21
"the known action of GC on dio2/dio3 can not explain the diversity of transcriptional responses in tailfin."
How are dio2/dio3 genes regulated after TH and/or CORT treatment in your experiment?
=> The following sentence was added at the end of the result section "Cross-talks do exist, and they only represent a fraction of transcriptional responses".
Importantly, we found dio2 to belong to the n_n_u cluster, meaning that its expression requires both T3 and CORT, whereas dio3 expression only depends on T3 (cluster u_n_u).
Results & Discussion
1) The authors should give the information of material (cultured tail explants or tailfins of tadpoles) in legends of Figs 2-4, and incubation time in legend of Fig. 4, especially about D.
=> Done
Legend Figure 2: "tailfin dissected from cultured tail explants" was added.
Legend Figure 3: "Only genes displaying consistent expression between explants culture and whole animals were kept. " added after the title of the figure.
Legend Figure 4: "Only transcriptional responses displaying consistent expression between explants culture and whole animals are considered. " added after the title of the figure. At section D, we added the incubation time "after 3 days treatments".
2) Letters are too small in size to recognize in Figs. 2-6.
=> Corrected. We nonetheless increased the size of all small characters by one to two font sizes.
3) The authors should explain the principal component analysis in Fig. 2A and the meaning of the character positions (T3, CORT, and T3-CORT) more in detail.
=> PCA is now a very standard quality control (normally) found in all RNA-Seq and functional genomics experiment. We believe that adding a tutorial or guidance about it would oversize its importance with respect to other analysis we carried out, and especially system biology, which is far less common despite its usefulness. In addition, a complete guide describing how to do it and how to interpret it is provided in the software package cited in the text (Anders et al., 2013). There is no real use to add it in the body of the manuscript.
PCA works as follows: it is a naïve (i.e. without prior assumption) multi-variate frameworks decomposing the total variance of a dataset into sets of technical and biological variability (or combinations of them) called principal components (PCs). The higher the variance of a component, the stronger its contribution in the datasets. Therefore, one should expect high quality experiments to associate high levels of variance with treatments, meaning that most of the variability reflect treatments. At the other extreme, low levels of variance associated with treatments imply a dataset dominated by noise. For interpretation, the datasets are located on a serie of scatterplots where axes represent principal components, and data points are projected to the corresponding axes (PC). Biological replicates should group together and samples segregate along PC1 and PC2 and form distinct groups. This is exactly the result shown Figure 2A.
4) The total number of DE genes is 729 in page 8, but this is 845 in page 10 (Fig. 3A). Why?
=> This effect is described in the section METHODS of the manuscript, in the section relative to the gene clustering method.
5) Almost all DE hubs involved in DNA methylation belong to T3 response type (Fig.4E), but 65.4% of DMRs are X-talks response type (Fig. 6A). Why?
=> This is exactly the point of our work: the uncoupling of the transcriptional response and DNA methylation dynamic. The detailed molecular details are not known, and we agree that this is intriguing. A first element of response is that many factors are involved in DNA repair-methylation. This accounts for hubs (gene products highly connected in the network) as well as gene products less connected. Part of the answer certainly lies in less connected genes (most of those shown Figure 4G). Intriguingly, a single hub is co-regulated by T3 and CORT (ADDITIVE response) and makes functional connections with DNA repair and epigenetics: the multi-task plateform UHRF2. Although we do not provide any additional data about this factor, it will certainly be worth examination in the future.
Also, we would like to pinpoint that our experiments measure changes of gene expression only, and many events may be regulated at other levels (post-translational modifications, alterations of intracellular Ca++ signaling…). Again, further experimental work is needed to draw a definitive answer.
But the referee is right; this point is intriguing and we added an entire section in the discussion:
The fact that all but one DE hubs acting on DNA methylation display T3 response might seem counter intuitive when DMR dynamics display mostly X-talks properties. Several line of evidence can help settle this apparent discrepancy. First, our experiments measure changes of gene expression and many events may be regulated at other levels (post-translational modifications, alterations of intracellular Ca++ signaling…). Second, even though UHRF2 is the only non-T3 hub (ADDITIVE) of the PPI network, it nonetheless makes a functional connection between DNA methylation and the transcriptional regulation by T3 and CORT. Furthermore, the functional response is certainly not limited to hubs and other non-hub factors may be involved. Two interesting candidates are OTUD4 and ZMPSTE24, for which the transcriptional response is also ADDITIVE (d_d_d and u_u_u, respectively). A quick BIOGRID survey indicates that in human, UHRF2 and OTUD4 engage physical interactions with USP7 (expressed in tailfin, but not DE), thus providing a good starting point to elucidate the molecular cascades involved.
6) 27.7% of DMRs overlap with, or are located within genes (Fig. 7A). Do these 3000 genes contain any genes involved in tailfin regression? If it is true, what are they? Which response type do they display?
=> Informations added in the results section "DMRs are located far from genes": "We note, however, that in our case, the repertoire of DE genes is very limited and corresponds to only five genes: B4GALNT4 (d_d_d), CHTF18 (d_n_d), PAPPA (n_n_u), ATP12A (n_u_n ) and ANGPTL2 (u_n_u)", followed by " This result is not surprising, as enhancers and response elements are expected to be located at large distances from their target gene, and even enhancers located within a gene may in fact regulate a different target"
7) p.14, line 15-18
"Results clearly show that most of them display increased or decrease expression relative to the non- treated control, and only a few genes display minute (or no) difference of expression (Figure 4G). "
"With only a few exceptions, gene response is very similar when treatments are applied to tail culture or whole animals. "
I cannot recognize them. Show the quantitative data to demonstrate them.
=> Although data are already available Table S2, we added a novel supplementary table (supp Table S5) containing the normalized read count and averaged over all biological replicates, for all genes involved in DNA methylation and corresponding to figure 4G.
8) There are many error candidates.
=> All fixed. We also reviewed the entire manuscript to edit typos.